# Non-genetic diversity modulates population performance

Adam James Waite[1,†] iD, Nicholas W Frankel[1,†,‡], Yann S Dufour[1,†,§] iD, Jessica F Johnston[1] iD,
Junjiajia Long[2] & Thierry Emonet[1,2,*] iD

## Abstract

Biological functions are typically performed by groups of cells that express predominantly the same genes, yet display a continuum of phenotypes. While it is known how one genotype can generate such non-genetic diversity, it remains unclear how different phenotypes contribute to the performance of biological function at the population level. We developed a microfluidic device to simultaneously measure the phenotype and chemotactic performance of tens of thousands of individual, freely swimming *Escherichia coli* as they climbed a gradient of attractant. We discovered that spatial structure spontaneously emerged from initially well-mixed wild-type populations due to non-genetic diversity. By manipulating the expression of key chemotaxis proteins, we established a causal relationship between protein expression, non-genetic diversity, and performance that was theoretically predicted. This approach generated a complete phenotype-to-performance map, in which we found a nonlinear regime. We used this map to demonstrate how changing the shape of a phenotypic distribution can have as large of an effect on collective performance as changing the mean phenotype, suggesting that selection could act on both during the process of adaptation.

**Keywords** cellular motility; chemotaxis; Jensen's inequality; non-genetic diversity; nonlinear systems
**Subject Categories** Microbiology, Virology & Host Pathogen Interaction; Quantitative Biology & Dynamical Systems; Signal Transduction
**Mol Syst Biol. (2016) 12: 895**

## Introduction

Biological functions are not typically carried out by isolated cells, but rather by populations of clonal or near-clonal cells that display a continuous distribution of phenotypes. In humans, for example, near-clonal T cells exhibit differences in migratory behavior (Baaten

*et al*, 2013), and clonal $\beta$ cells in the pancreas display different insulin thresholds (Benninger & Piston, 2014). Clonal cancer cells display variation in resistance to chemotherapeutics (Fallahi-Sichani *et al*, 2013; Flusberg *et al*, 2013; Kreso *et al*, 2013). In microbes, non-genetic diversity is implicated in biofilm formation (Høiby *et al*, 2011), virulence (Ackermann, 2015), and antibiotic persistence (Bigger, 1944; Balaban *et al*, 2004).

While the ways in which one genotype can give rise to a distribution of phenotypes are relatively well understood (Levin *et al*, 1998; Elowitz *et al*, 2002; Raser & O'Shea, 2004; Colman-Lerner *et al*, 2005; Jones *et al*, 2014), the ways in which non-genetic diversity modulates population performance remain unclear. The difficulty is that the ability of a clonal population to perform a biological function emerges as a convolution of the distribution of phenotypes in the population, $P(X)$ (Fig 1A, left), with the function that relates individual phenotype to performance, $\varphi(X)$ (Fig 1A, right), where $X$ is a random variable describing the phenotype. An important consequence of this convolution is that if $\varphi(X)$ is nonlinear, the population performance can become very sensitive to non-genetic diversity, since outliers in the distribution of $P(X)$ (Fig 1A, left, bright pink region) that display nonlinear performance characteristics (Fig 1A, right, bright green region) may have disproportionate effects on population performance (Golowasch *et al*, 2002). Thus, determining the relationship between non-genetic diversity and population performance requires knowing the full distribution of $P(X)$ and the functional form of $\varphi(X)$. Although it is possible to accurately determine $P(X)$ using flow cytometry, microscopy, and microfluidics, determining $\varphi(X)$ is difficult because it requires simultaneous measurement of both phenotype and performance in the same individual.

To overcome this difficulty, we developed a microfluidic device (Fig 2A) that enabled us to determine the swimming phenotype and chemotactic performance of a large number of individual *Escherichia coli* in the presence of a stable gradient of the non-metabolizable chemoattractant α-methylaspartate (MeAsp; Fig EV1). *E. coli* chemotaxis provides an ideal model of non-genetic diversity because the pathway is well understood, and genetically identical *E. coli* cells exposed to a uniform environment display a continuous

1   Department of Molecular, Cellular, and Developmental Biology, Yale University, New Haven, CT, USA
2   Department of Physics, Yale University, New Haven, CT, USA
    *Corresponding author. Tel: +1 203 432 3516; E-mail: thierry.emonet@yale.edu
    †These authors contributed equally to this work
    ‡Present address: Department of Cellular and Molecular Pharmacology, University of California San Francisco, San Francisco, CA, USA
    §Present address: Department of Microbiology and Molecular Genetics, Michigan State University, East Lansing, MI, USA

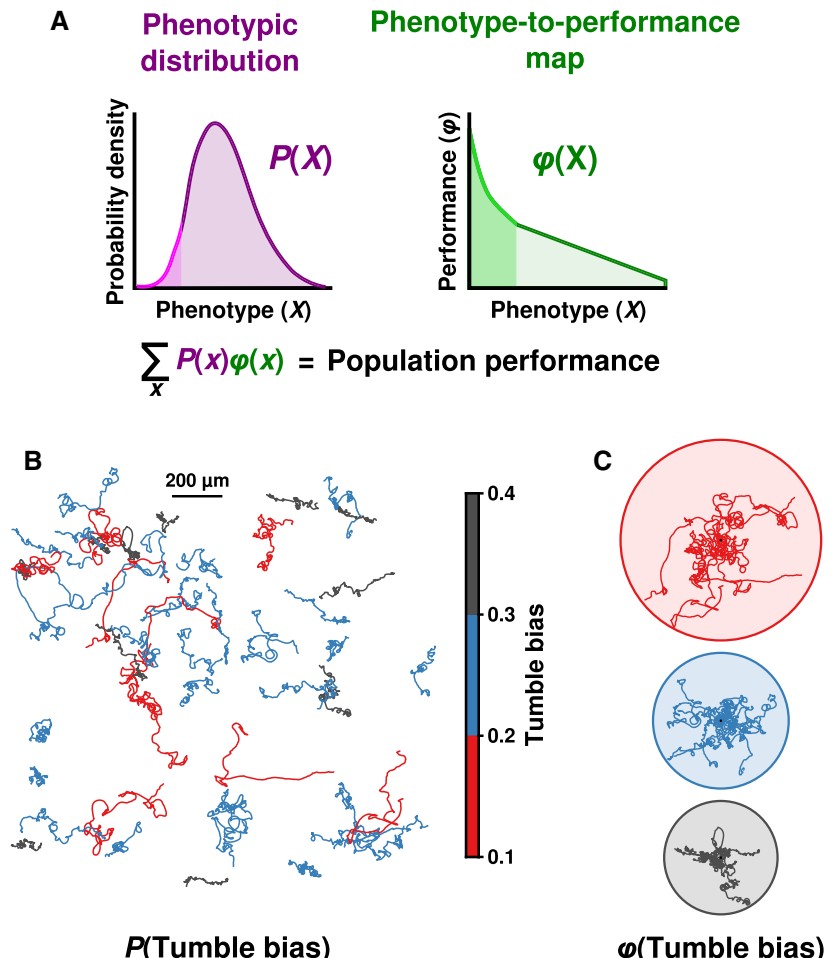

**Figure 1. Population performance is a convolution of non-genetic diversity and the relationship between phenotype and performance.**

A   The convolution of non-genetic diversity, *P(X)*, with the function relating phenotype to performance, *φ(X)*, determines population performance. A nonlinear region of *φ(X)* (bright green) may enhance the performance of rare phenotypes (light magenta), which would affect population performance in non-intuitive ways.

B   2D trajectories of clonal *Escherichia coli* cells in a uniform environment display differences in tumble bias. Cells were placed on a glass slide and tracked for 1 min.

C   Cells from (B) grouped by tumble bias with all starting positions at same point (black dot; ten randomly chosen tracks per circle). Circles indicate the maximum extent traveled by the phenotype.

distribution of swimming phenotypes (Fig 1B; Spudich & Koshland, 1976; Park *et al*, 2010; Dufour *et al*, 2016).

Clonal cells express the same motility genes and swim by alternating between relatively straight "runs" and direction-changing "tumbles". However, the probability that a cell is tumbling (its "tumble bias") varies from cell to cell (Park *et al*, 2010). When cells detect an increase in the concentration of a chemical attractant via signaling from transmembrane receptors, they transiently decrease their tumble bias, which extends runs up the gradient. This results in biased motion in the desired direction (Sourjik & Wingreen, 2012). An adaptation mechanism returns cells to their pre-stimulus tumble bias and increases the range over which cells can sense changes in attractant concentration by several orders of magnitude (Sourjik & Wingreen, 2012). The characteristic timescale of this adaptation, or "adaptation time", also varies from cell to cell (Spudich & Koshland, 1976; Park *et al*, 2010). Together, tumble bias and adaptation time, which are inversely correlated with one another (Alon *et al*, 1999; Park *et al*, 2010; Frankel *et al*, 2014), account for a substantial

amount of a cell's swimming behavior (Emonet & Cluzel, 2008; Vladimirov *et al*, 2008; Frankel *et al*, 2014; Dufour *et al*, 2016). Here, we use tumble bias as the phenotype in our assays, since measuring adaptation time typically requires immobilizing cells (Alon *et al*, 1999; Min *et al*, 2012; but see also Masson *et al*, 2012) and precludes simultaneously measuring their swimming performance.

Because cells displaying lower tumble bias spend more time running, their motion is more diffusive and exploratory than cells that have higher tumble bias (Fig 1C; Berg, 1993; Dufour *et al*, 2016). Previous modeling work predicted that these different tumble bias phenotypes would perform differently in the same environment (Dufour *et al*, 2014; Frankel *et al*, 2014) and that mutations altering regulatory elements in the chemotaxis pathway could have profound consequences for population gradient-climbing performance (Frankel *et al*, 2014). However, there is currently no experimental evidence to support the idea that the non-genetic diversity observed in clonal populations of *E. coli* is enough to influence population functional performance. Experimental evidence in

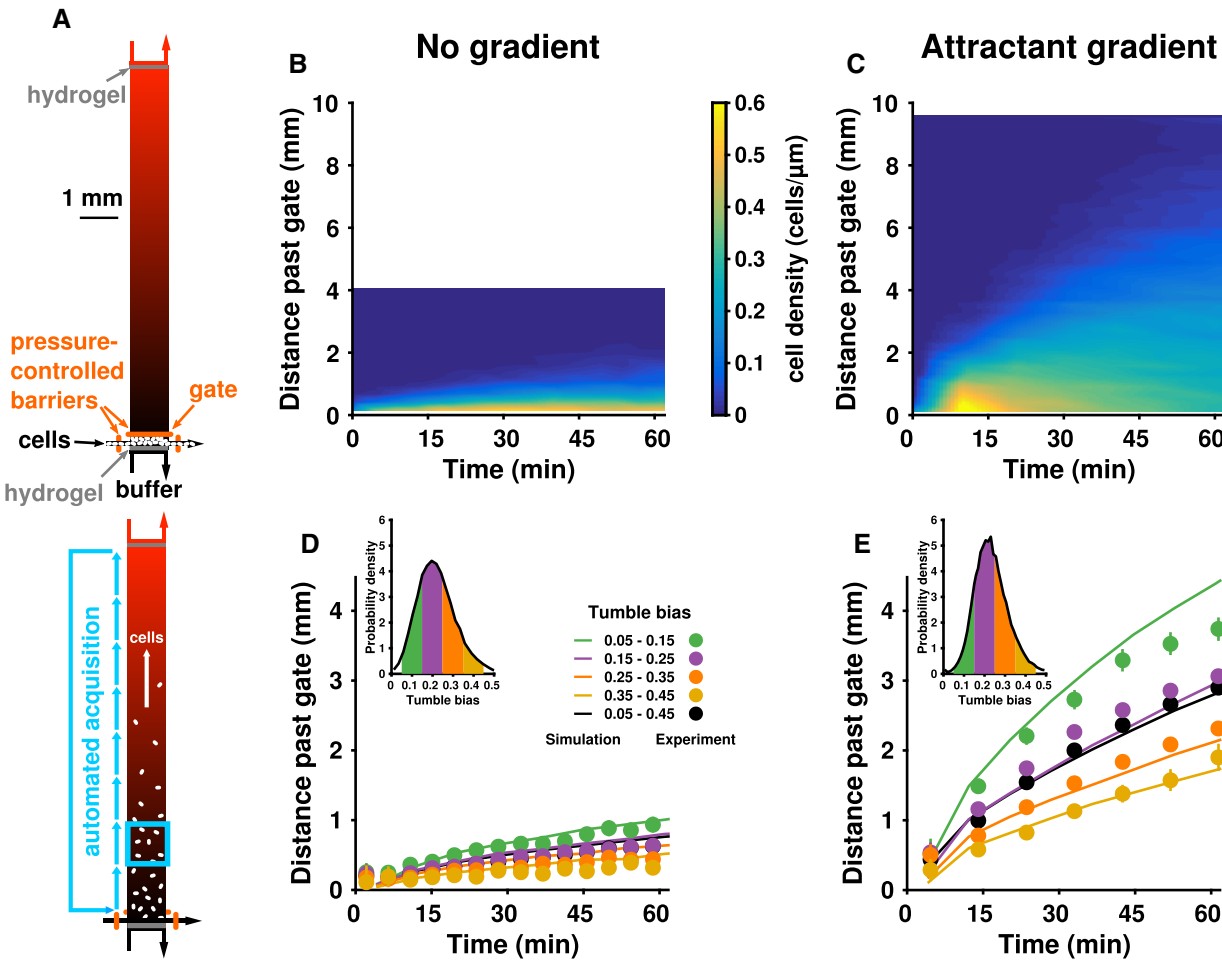

**Figure 2.  Non-genetic diversity results in spatial segregation and differential performance of phenotypes during diffusion and drift.**

A     Schematic diagram of the microfluidic device before (top) and after (bottom) opening the gate (horizontal orange bar). Once the gate was lifted, cells were allowed to explore the chamber and perform chemotaxis in a stable gradient during which short movies (light blue box) were acquired. See text for details.

B, C   Kymographs showing the density of cells at each position and time recorded in the movie in the absence (B) or presence (C) of a 0.1 mM/mm gradient of the non-metabolizable attractant, α-methylaspartate (MeAsp). Each movie was split into five regions to calculate density across the frame. The resulting data were then linearly interpolated to produce the kymograph. White space indicates areas that were not observed.

D, E   After determining the tumble bias of each trajectory, the population was split into four equally spaced tumble bias bins (insets), and the mean position of each tumble bias bin was plotted over time (filled circles, experimental measurements; lines, theoretical simulations) in experiments without (D) and with (E) a gradient of MeAsp.

Data information: Data for (B and D) were combined from two independent experiments, totaling 11,100 cells. Data for (C and E) were combined from four independent experiments, totaling 33,400 cells. Simulation data were obtained from 16,000 cells. All error bars show ± two times the standard error of the mean.

support of this idea would further the larger hypothesis that non-genetic diversity in cellular populations is an adaptive trait.

We found that spatial structure rapidly emerged along the gradient with cells at the leading front drifting up the gradient much faster than those at the back. We traced the origins of this spatial structure to non-genetic diversity, which causes "un-mixing" of different tumble bias subpopulations over time. This separation was not solely due to the random dispersal inherent in run-and-tumble motility, but rather to the fact that different phenotypes performed differently—that is, drifted at different rates—in the same environment. In order to understand how the distribution of tumble bias phenotypes gave rise to population performance, we manipulated this distribution by perturbing gene expression,

and this altered population performance. By combining experiments at different protein expression levels, we empirically constructed $\varphi(X)$, the map that links tumble bias to chemotactic performance. Significantly, we found that this relationship is convex at low values of tumble bias, leading to disproportionate performance of populations containing low-tumble-bias cells. We show that this convex relationship results in "Jensen's inequality" (Jensen, 1906) for populations containing low-tumble-bias cells: Population performance is greater than the performance of the mean population phenotype. Using $\varphi(X)$, we predicted the performance of arbitrary tumble bias distributions and showed how the shape of $P(X)$ becomes an important determinant of population performance for nonlinear regimes of $\varphi(X)$.

# Results

We used our microfluidic device (Fig 2A) to see how different phenotypes within an isogenic population of *E. coli* would distribute themselves over space and time. To prepare cells for the assay, they were grown in minimal media, harvested at mid-exponential phase, washed, and resuspended in a motility buffer that does not allow growth. They were then loaded into the device at a low density to minimize cell-to-cell interactions (Materials and Methods). Cells were initially confined to a small (0.3 mm long by 1 mm wide) region at one edge of the device (Fig 2A, top) by using a hydraulic gate (Unger *et al*, 2000; Materials and Methods). At the beginning of the experiment, we lifted the gate to conduct a bacterial "race" through the observation chamber (dimensions: 10 mm long × 1 mm wide × 10 μm deep). Performance was defined as the distance traveled past this gate (Fig 2A, bottom). We successively acquired short (1 min) movies along the length the device for 1 h. In each movie, we tracked individual cells (Materials and Methods). Then, we determined the tumble bias of each trajectory using a probabilistic model that classified every time point along a trajectory as either a run or a tumble based on information about cell velocity, acceleration, and angular acceleration (Materials and Methods; Dufour *et al*, 2016). We verified that cells in the device exhibited a distribution of tumble bias similar to that measured using tethered cells (Park *et al*, 2010) and remained stable for at least an hour (Appendix Fig S1). This observation confirmed that single clonal *E. coli* cells differ in their swimming phenotype and prompted us to use our device to determine the consequences of this variability for population performance.

In the absence of a gradient of MeAsp, cells explored the observation chamber through passive diffusion (Fig 2B). The farthest they traveled was ~1 mm past the gate. We compared this to performance in a 0.1 mM/mm gradient of MeAsp, which we created using circulation channels adjacent to the near and far side of the chamber (Fig 2A; Materials and Methods). In this case, cells traversed the chamber much farther than in the diffusion experiment (Fig 2C), demonstrating that the device was able to specifically elicit and measure chemotaxis with single-cell resolution. Surprisingly, some cells were visible at the source side of the chamber, over 5,000 cell-lengths away (~10 mm), in less than 45 min (Figs 2C and EV2). This demonstrated wide variation in gradient-climbing performance within a clonal population and suggested the presence of high-performance subpopulations.

We evaluated the previous theoretical prediction that cells with different tumble bias should perform differently (Dufour *et al*, 2014; Frankel *et al*, 2014) by categorizing trajectories into four equally spaced ranges of tumble bias (insets of Fig 2D and E for without and with a MeAsp gradient, respectively). The mean positions of these subpopulations over time showed that the different subpopulations traveled across the chamber at different rates (Fig 2D and E, filled circles; black circles show the total population mean). In both the absence (Fig 2D) and presence (Fig 2E) of a MeAsp gradient, lower tumble bias subpopulations traveled at a faster rate through the chamber than higher tumble bias subpopulations. This was more obvious in the MeAsp gradient, where the lowest tumble bias subpopulation (Fig 2E, green circles) outperformed the population average (Fig 2E, black circles) by ~500 cell-lengths (~1 mm) after 45 min (see also Fig EV2). These differences

in performance could not be explained by differences in cell speed alone (Appendix Fig S2).

To verify that differences in phenotype were sufficient to produce this spatial separation effect, we used our model (Frankel *et al*, 2014) to produce stochastic simulations of individual cells. In our model, phenotypic variability in the functional parameters of chemotaxis (adaptation time, tumble bias, and noise in the adaptation mechanism) could only arise from cell-to-cell differences in the levels of chemotaxis proteins. Cell-to-cell differences in protein levels were determined by an experimentally constrained model of noisy gene expression. Cell-to-cell differences in run speed, which were modeled as coming from a normal distribution and were uncorrelated with the other phenotypic parameters, contributed additional variation (Materials and Methods). We adjusted gene expression to match the experimentally observed distribution of tumble bias without a MeAsp gradient (Fig EV3A). These parameters were also used for the experiments with a gradient (Fig EV3B). Small differences in observed speed necessitated manually matching the speed distributions separately to the diffusion and drift experiments (Fig EV3, right column). All other parameters were kept the same. The trajectories output by each simulation were processed as if they were observed by our microscope's acquisition procedure (Materials and Methods). Simulation results largely agreed with experimental results (Fig 2D and E; compare lines to points), demonstrating, as previously predicted (Dufour *et al*, 2014), that, in a liquid environment, low-tumble-bias cells outperform other subpopulations. Overall, the quantitative agreement between experiment and simulations validated our model, which was able to predict not just the average population performance, but also that of the individual phenotypes. These simulations confirmed that non-genetic diversity was sufficient to produce the performance differences that lead to population separation in this environment.

Since the experimental gradient was approximately linear, and the chemosensory receptors are log-sensing, cells likely perceived the gradient as becoming shallower as they climbed it (Fig EV1). Thus, the measured tumble bias of individual cells was predicted to change somewhat over time, with cells experiencing larger transient drops in tumble bias at the beginning of the experiment than in the relatively shallow gradient farther along the chamber. This deviation could have resulted in our underestimation of cells' steady-state, or unstimulated, tumble bias, especially at earlier time points. Since we could not directly measure the unstimulated versus stimulated tumble bias of individual cells in our experimental setup, we used our model to perform simulations in the experimentally derived gradient. In these control simulations, all cells were initialized with the same internal state. We found that the difference between stimulated and unstimulated tumble bias was reduced to a small, negative value (−0.004) within the first millimeter of the device (Appendix Fig S3A) and could not account for the observed differences in performance (Appendix Fig S3B). Therefore, in our experiments, any difference between the observed and unstimulated tumble bias was likely negligible after the first millimeter, which was less than the distance covered by an individual movie.

The superior performance of low-tumble-bias phenotypes in liquid media prompted us to understand these phenotypes in greater detail. However, in our experiments, low-tumble-bias phenotypes were relatively rare in the "wild-type" strain RP437. This was probably due to its history of laboratory selection on soft-agarose plates,

                                                     

since obstacles in soft agarose are known to select against low-tumble-bias cells (Armstrong *et al*, 1967; Wolfe & Berg, 1989). We therefore sought to modify the population distribution using information from our model, which predicted that population performance could be shaped by manipulating the level of expression of specific chemotaxis proteins.

The well-known architecture of the two-component chemotaxis signaling pathway (Fig 3A) suggested that tumble bias could be altered by modifying the expression of chemotaxis proteins related to either the activity of the response regulator (CheY and CheZ) or the adaptation time mechanism (CheR or CheB). We initially chose to change the expression level of CheR because doing so preserves the experimentally measured inverse correlation between tumble bias and adaptation time (Alon *et al*, 1999; Park *et al*, 2010). We did not alter CheB because it is involved in an additional phosphory-lation feedback loop (Kollmann *et al*, 2005; Sourjik & Wingreen, 2012) which could have complicated the interpretation of the results. Changes in expression of CheR are known to affect tumble bias and chemotactic behavior on average (Alon *et al*, 1999; Løvdok *et al*, 2009; Park *et al*, 2011), but how these changes affect non-genetic diversity and how this diversity changes the resulting population performance are less well understood. While recent theoretical work has predicted a causal relationship between individual gene expression, chemotactic phenotype, and population performance (Frankel *et al*, 2014), we sought to experimentally test these predictions.

In order to create a suitable strain to study these relationships, we took a Δ*cheR* derivative of RP437 and integrated a single-copy, mCherry-tagged version under inducible control of the native chromosomal *lac* operon (Materials and Methods). Using this mutant, we created two populations with different average expression levels of CheR by growing cultures in the presence of low (10 μM) or high (100 μM) concentrations of isopropyl-β-D-1-thiogalactopyranoside (IPTG). When cells were induced with a high concentration of IPTG, CheR expression was higher overall (Fig EV4A, blue line) and the tumble bias distribution was comparable to that of wild-type cells (Figs 3B and EV4B). Cells induced with a low concentration of IPTG had lower expression of CheR overall (Fig EV4A, red line) and displayed fewer tumbles during their observation (Figs 3C and EV4B). Importantly, the minimum measurable tumble bias in our experiments was ~0.002. This means that any cells with a tumble bias greater than 0 but less than 0.002 were recorded as having an observed tumble bias of 0. These results demonstrate a connection between chemotaxis protein expression level and tumble bias on the population level, which was recently demonstrated at the single-cell level using a similar artificial expression system (Dufour *et al*, 2016). To account for the preponderance of lower tumble bias individuals in the low-CheR population, we added two more tumble bias categories to our analysis, 0–0.005 (0 or 1 tumbles/min) and 0.005–0.05 (2–26 tumbles/min). In the wild-type and high-CheR mutant populations, there were not an appreciable number of cells observed in these categories even with the large number of trajectories measured in each experiment, so these categories were not used.

Having generated a new distribution of phenotypes (Fig 3C), we modeled the high- and low-CheR populations by adjusting the cell run speed and the mean expression level of CheR to match the experimentally observed distributions (Fig EV3C and D). We left all other parameters unchanged. Theory predicted that, in a liquid environment, the rate at which a cell climbs the gradient should be a non-monotonic function of tumble bias (Dufour *et al*, 2014). As tumble bias decreases, performance should increase, until tumble bias approaches zero. At this point, performance should decline abruptly, since cells that never tumble can no longer perform chemotaxis (Parkinson, 1978). Simulations of the populations generated from the mutant strain predicted that cells in the lowest tumble bias category (0–0.005) would climb the gradient much more rapidly than the other categories (Fig 3D, red dashed line), which was in agreement with our experimental results (Fig 3D, red triangles). The performance advantage of the lowest tumble bias bin would likely have been even greater if the chamber were longer, but, due to the finite length of the device, the increase in mean position of the lowest tumble bias cells began to saturate after only ~15 min of observation due to accumulation of cells at the attractant source (Fig EV2). This gradual slowing is also seen in the simulations, which take into account the dimensions of the device and observation region (Fig 3D, red dashed line). Because the low-CheR population was dominated by these very low-tumble-bias cells, the mean distance traveled by the low-CheR population was also greater than the mean distance traveled by the high-CheR population (Fig 3F, compare filled triangles to open circles). Both populations outperformed an uninduced population (Fig 3F, open squares), as well as a non-chemotactic (Parkinson, 1978) *cheY* mutant that only runs (Fig 3E and F, x's; see Fig EV2 for corresponding cell density profiles). Thus, as predicted (Dufour *et al*, 2014), performance increased dramatically with decreasing tumble bias, but then decreased just as dramatically when the tumble bias approached zero. Overall, simulations predicted the behavior of all phenotypes (compare lines to points in Figs 2D and E, and 3D). These results experimentally demonstrated a causal relationship between gene expression, phenotype, and performance and showed that the model was able to predict cell behaviors even outside the wild-type range of protein expression.

As mentioned earlier, adaptation time and tumble bias are inversely correlated in wild-type cells, and changing CheR maintains this correlation. We therefore wondered how performance would be affected if tumble bias alone was modified. To manipulate tumble bias without changing adaptation time, we performed experiments on an RP437-derived Δ*cheYcheZ* strain containing IPTG-inducible CheY tagged with mRFP and arabinose-inducible CheZ tagged with EYFP on plasmids (Fig EV5A; Sourjik & Berg, 2002). We adjusted the CheY-to-CheZ ratio (20 μM IPTG, 0.001% arabinose) to produce a tumble bias distribution with a low mean (Fig EV5B) and found that, as with the inducible CheR strain, lower tumble bias resulted in higher performance (Fig EV5C). The inducible CheY/CheZ strain had a greater run speed than the inducible CheR strain (34 μm/s versus 21 μm/s), causing the inducible CheY/CheZ cells to move through the chamber faster and reach the end of the device sooner than the inducible CheR cells. The high performance of these low-tumble-bias cells was consistent with a recent theoretical prediction: For cells in shallow gradients, changing tumble bias should have a greater effect on performance than changing adaptation time for a wide range of tumble bias (see Fig 2A in Dufour *et al*, 2014).

Taken together, the experimental data indicated that the relationship between tumble bias and gradient-climbing performance was

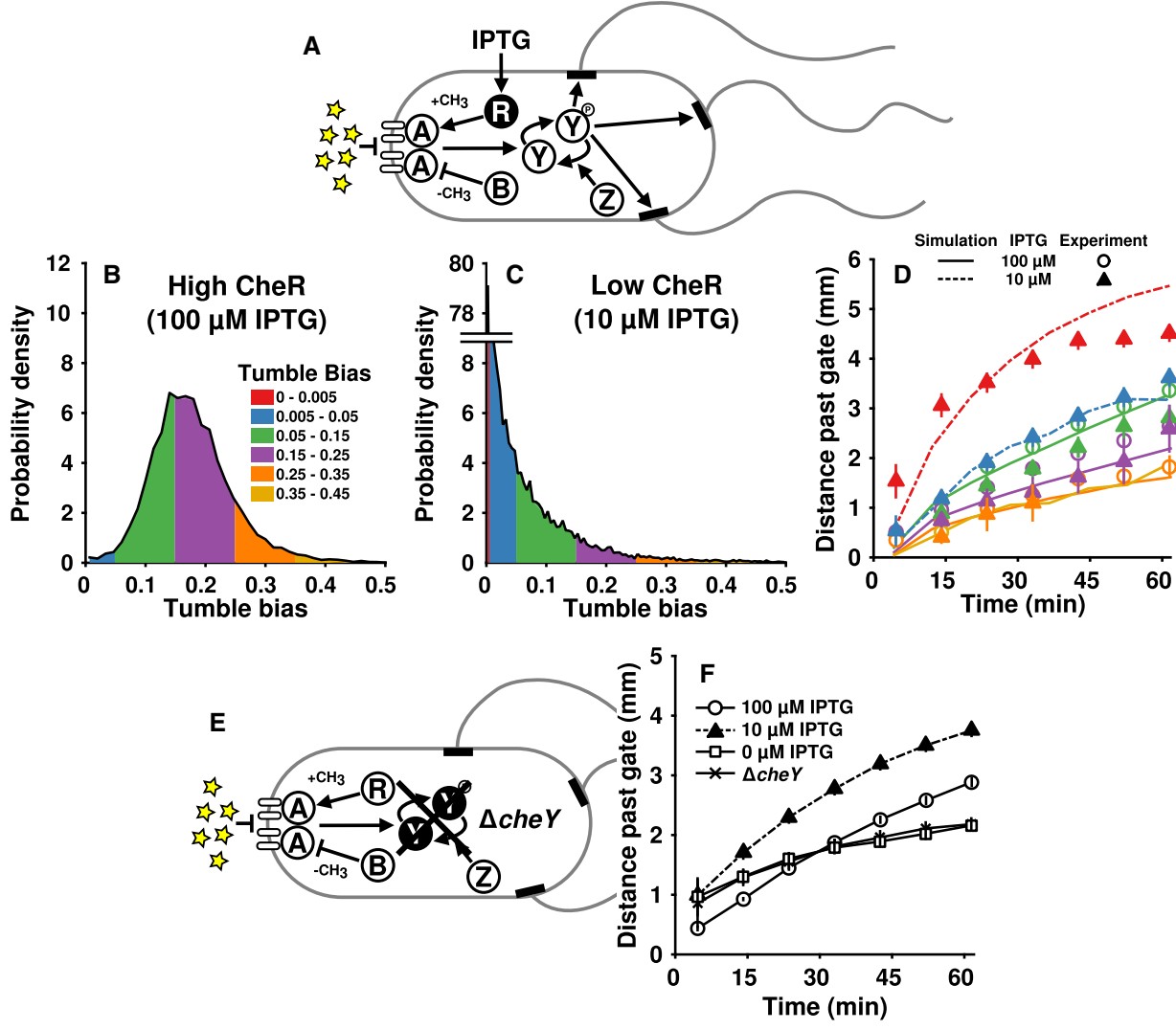

**Figure 3.  Expression level of a single chemotaxis protein alters tumble bias distribution and gradient-climbing performance.**

A       Diagram of the chemotaxis pathway in *Escherichia coli*. We constructed a mutant strain with IPTG-inducible expression of *cheR* (black). Chemoattractant (yellow stars) interacts with receptors (open ovals) to suppress the activity of CheA (A). CheA phosphorylates CheY (Y). Phosphorylated CheY (Y-p) diffuses to the motors (black rectangles) inducing switching from counterclockwise (CCW) to clockwise (CW) rotation. A switch from CCW to CW disrupts the flagellar bundle, which causes a tumble. CheR (R) increases and CheB (B) decreases the sensitivity of the receptors via methylation and demethylation, respectively.

B, C    Tumble bias distributions of cell populations in the microfluidic device with a gradient after induction with 100 μM (B) or 10 μM (C) IPTG.

D       The mean position over time in the chamber for populations induced with 100 μM (open circles) or 10 μM (filled triangles) IPTG. Lines were generated from simulations of populations with tumble bias and speed distributions matched to the data.

E       Cells lacking CheY (black) cannot perform chemotaxis because motors without CheY-p activation are permanently in the counterclockwise (run) state.

F       The mean performance of 100 μM IPTG (open circles), 10 μM IPTG (filled triangles), 0 μM IPTG (open squares), and Δ*cheY* (x's). Lines are guides for the eye. Two experiments were combined for each of the 100, 10, and 0 μM IPTG populations, totaling 12,900, 15,300, and 11,600 cells, respectively. Two experiments (totaling 8,640 cells) were combined for the Δ*cheY* data. All simulated data were obtained from 16,000 cells per population. Only points with at least 45 cells were plotted.

Data information: All error bars show ± two times the standard error of the mean.

nonlinear and non-monotonic: Performance increased steeply as tumble bias decreased, making the overall shape convex, and it decreased as tumble bias approached zero. The nonlinear regions of this relationship mathematically implied that population performance may not equal the performance of the average phenotype. This is formalized by Jensen's inequality, which states that, for a convex function $\varphi(X)$ of a random variable $X$, $E[\varphi(X)] \geq \varphi(E[X])$, where $E[X]$ is the expected value, or mean, of $X$ (Jensen, 1906). (For a

concave function, the inequality is reversed.) In our case, $\varphi_t(TB)$ was the performance (distance traveled) of phenotype (tumble bias) $TB$ evaluated at time $t$. The distinction between the performance of the population and the performance of its mean phenotype is potentially important because it suggests that, if a nonlinear performance regime is present, a population with non-genetic diversity may perform differently than expectations based on its population-averaged phenotype.

To investigate this possibility, we created an empirical phenotype-to-performance map, $\varphi_t(TB)$, over a large range of tumble bias by combining the data from the experiments performed with the inducible CheR strain. Consistent with our earlier impression, we found that $\varphi_t(TB)$ was convex at low (but nonzero) tumble bias and relatively linear at high tumble bias (Fig 4A). Given the existence of a convex region, Jensen's inequality predicted that the mean performance of the population should be strongly affected by the presence of low-tumble-bias phenotypes. Specifically, in the low-CheR population, we expected the mean performance of the population, $E[\varphi_t(TB)]$, to be greater than the performance of the mean phenotype, $\varphi_t(E[TB])$, due to the disproportionately high performance of low-tumble-bias cells. In the high-CheR population, we expected these differences to be smaller since in this region $\varphi_t(TB)$ was linear.

These predictions were confirmed by the data. For the low-CheR population, $E[\varphi_t(TB)] > \varphi_t(E[TB])$ for all time points (Fig 4B), while for the high-CheR case (Fig 4C), $E[\varphi_t(TB)] \approx \varphi_t(E[TB])$ for all time points. Although less obvious due to the cells' higher speed and consequently faster equilibration throughout the device, Jensen's inequality also appeared to apply to cells with low CheY at early time points (Fig EV5D). This suggested that the nonlinear portion of $\varphi_t(TB)$ was not solely the result of the increased adaptation time of the low-CheR cells.

The functional form of the phenotype-to-performance map, $\varphi_t(TB)$, suggested that there should be at least two ways in which selection could alter the non-genetic diversity of a population to improve its performance. One way would be to alter the mean phenotype, $E[TB]$, without changing the shape of the phenotypic

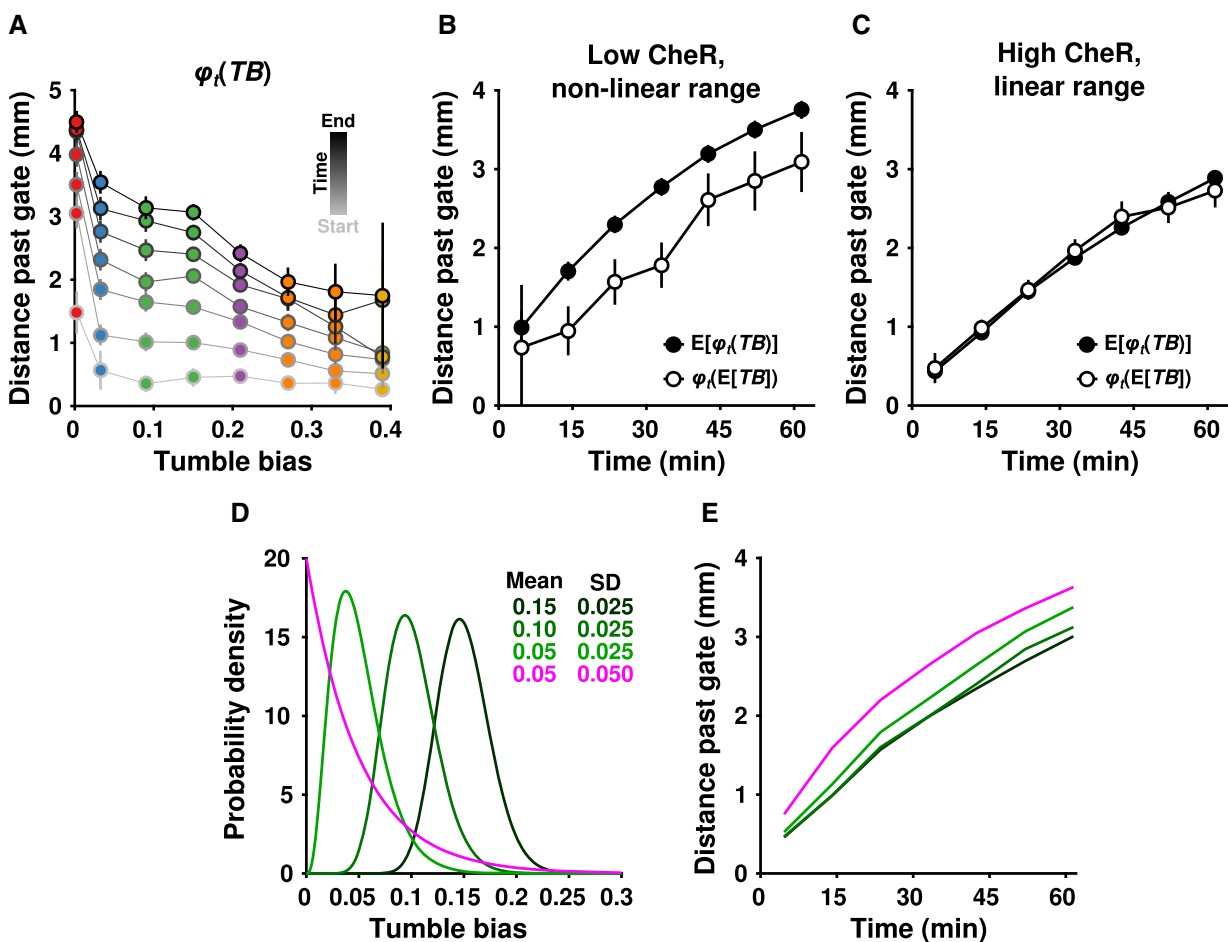

**Figure 4. Non-genetic heterogeneity strongly affects mean population performance when performance is a convex function of phenotype.**

A    Data from the 100 µM and 10 µM IPTG experiments were combined to produce a map from phenotype (tumble bias, *TB*) to performance (distance past the gate, $\varphi_t(TB)$). The lowest tumble bias point is 0.0025.

B, C    The mean performance of the population (closed circles, same as in Fig 3F) and the performance of the mean phenotype (open circles) over time for cells induced with 10 µM (B) or 100 µM (C) IPTG. The mean tumble bias ± 2 standard errors was 0.044 ± 0.0005 for the low-CheR population (B), and 0.175 ± 0.0006 for the high-CheR population (C). In each case, the performance of the mean phenotype was calculated by averaging the performance of cells within 0.01 of the mean tumble bias. In all cases, error bars indicate ± two times the standard error of the mean.

D, E    Predictions of performance for hypothetical populations. (D) A series of hypothetical tumble bias distributions were generated using a gamma function with parameters [$k; \theta$] of [36; 4.17 × 10$^{-3}$], [16; 6.25 × 10$^{-3}$], [4; 0.0125] (dark to light green), and [1; 0.05] (magenta). The parameters for the green curves were chosen to have decreasing means and identical standard deviations (SDs). The light green and magenta distributions were chosen to have the same mean and different SDs. (E) $\varphi_t(TB)$ from (A) was interpolated and used to generate predicted performance curves for the tumble bias distributions in (D).

     

distribution $P(TB)$. Another way would be to alter the shape of $P(TB)$ without changing $E[TB]$. For example, through mutations in regulatory elements that changed protein expression noise (Frankel *et al*, 2014) or mutations that affected protein partitioning during cell division. Since the empirical map $\varphi_t(TB)$ describes the performance of any phenotype, we sought to examine these possibilities by creating different hypothetical populations $P(TB)$ and convolving them with $\varphi_t(TB)$ (Fig 4D and E, following the concept in Fig 1A). For simplicity, we used a gamma distribution for $P(TB)$. When populations contained cells that were mostly in the linear regime of $\varphi_t(TB)$, reducing the mean phenotype from 0.15 to 0.10 had a negligible effect (Fig 4E, compare dark to medium green). Reducing it further to 0.050 put more low-tumble-bias cells in the nonlinear regime, significantly increasing population performance (Fig 4E, light green). However, an even larger improvement in performance was obtained when the mean was maintained at 0.050, but the shape of the distribution was widened to include more cells in the nonlinear region (Fig 4E, magenta). This occurred because the nonlinear performance increase provided by the low-tumble-bias cells more than compensated for the presence of the poorer-performing high-tumble-bias cells. Thus, depending on the functional form of $\varphi(X)$, changes to the shape of the non-genetic diversity, $P(X)$, can have large effects on population performance, on a par with, or greater than, changes to the mean, $E[X]$, alone.

## Discussion

We have shown that non-genetic diversity observed in *E. coli* swimming behavior determines population performance and can be directly modulated by changes in the expression of individual genes. Critically, we found that changing gene expression modifies not only the mean performance of a population, but also the discrepancy between the mean performance and that of the mean phenotype. In other words, changing gene expression can affect how important the tail of the distribution of phenotypes is in determining population performance. This is because such changes can place the population in a regime in which the relationship between phenotype and performance is nonlinear. In *E. coli*, it was possible to enter this nonlinear regime by decreasing tumble bias through modulating the expression level of proteins relating to either the adaptation mechanism (CheR; Fig 3) or the response regulator (CheY and CheZ; Fig EV5). This confirms previous theoretical work which showed that tumble bias could have a large influence on performance (Dufour *et al*, 2014). Because biological function is typically executed by many cells expressing the same pathway, these results provide quantitative insight into the ways in which non-genetic diversity modulates biological function.

The implications of our findings are most immediately relevant to pathogenic strains of *E. coli* and other pathogenic bacteria such as *Salmonella enterica* (which shares an almost identical chemotaxis pathway with *E. coli*), which require chemotaxis for infection (Stecher *et al*, 2013). Characterizing population non-genetic diversity would be crucial, for example, in predicting the arrival time of bacteria to a wound site and the subsequent establishment of an infection, which often requires < 100 pathogenic organisms (Sewell, 1995). In this scenario, equating population performance with its mean performance would lead to serious overestimates of the initial

time of arrival due to the nonlinear impact of high-performing subpopulations on the performance of the entire population. In other scenarios, such as those requiring a large group of cells to arrive in a location precisely at the same time, or for cells to stay close to an attractant source, non-genetic diversity might hinder successful completion of the task. In such cases, we would expect the phenotype-to-performance map to be concave rather than convex (Frankel *et al*, 2014), favoring the reduction of diversity through selection.

Our data demonstrate the direct biological relevance of Jensen's inequality for biological behavior, as well as the fundamental impact of nonlinear performance curves on collective performance in heterogeneous populations. Although in this case the performance function was convex, our results can be easily generalized to both convex and concave nonlinearities in performance, as mathematically both are included under this formalism. Since convex and concave relationships between phenotype and performance (or fitness) are a common feature of biological systems (Golowasch *et al*, 2002; Simonsen *et al*, 2014; Pickett *et al*, 2015; Sinclair, 2015), Jensen's inequality is likely to apply to many other populations. In such cases, the conventional concept of "average phenotype" (i.e., one which is provided by population-averaged measurements) will provide a poor prediction of performance; rather, the performance distribution of diverse individuals in genetically identical populations must be considered.

It was previously shown that mutations in regulatory elements can alter the variability of gene expression as measured by mRNA transcripts (Jones *et al*, 2014), raising the possibility that gene expression noise is an evolutionarily tunable parameter. Here, we build on these results by linking the phenotypic distribution of a population to its performance of a specific task and highlight the effect of nonlinearity in the performance function. Since natural selection acts on phenotypes based on their performance, we have established the existence of the necessary feedback between gene expression and the environment that makes selection on non-genetic diversity plausible. Together, these mechanisms provide experimental support for the idea (Frankel *et al*, 2014) that a clonal population might be able to adapt to complex, time-varying environments via mutations in regulatory elements, thereby achieving higher fitness while bypassing potentially deleterious modifications to existing protein-encoding genes.

## Materials and Methods

### Strains and media

Unmodified RP437 was used as the "wild-type" strain. The inducible CheR strain was based on RP437; details of its construction are given below. The IPTG-inducible, mRFP-tagged CheY, arabinose-inducible, EYFP-tagged CheZ strain was based on a ΔcheYcheZ derivative of RP437 (Sourjik & Berg, 2002). The *cheY*-containing plasmid was pTrc99A, which provided ampicillin resistance, while the *cheZ*-containing plasmid was pBAD33, which provided chloramphenicol resistance. Cells were grown in M9–glycerol medium (M9 salts [12.8 g/l Na$_2$HPO$_4$, 3.0 g/l KH$_2$PO$_4$, 0.5 g/l NaCl, 1.0 g/l NH$_4$Cl], 1.0 g/l tryptone, 10 g/l glycerol). Chemotaxis buffer (CB; M9 salts with 0.1 mM EDTA, 0.01 mM L-methionine, and 10 mM

DL-lactate, 0.05% w/v polyvinylpyrrolidone-40) was used for washing cells and in the microfluidic device.

## Microfluidic device design and fabrication

Microfluidic devices were constructed from the biocompatible and oxygen-permeable silicone polymer polydimethylsiloxane (PDMS) on coverglass following standard soft lithography protocols for two-layer devices (Xia & Whitesides, 1998). The master molds for the device consisted of two silicon wafers with features created using ultraviolet (UV) photoresist lithography. The bottom wafer had features for the device channels and was created using SU-8 negative resist. Portions of the channel layer that were designed be opened and closed by pressure actuated valves were created with a second coat of SPR positive resist on the same wafer, which was used to create a rounded channel profile that can collapse fully if depressed from above. The second, top wafer contained features for the control channels that close the collapsible features in the bottom wafer. The top wafer was created using SU-8 negative resist.

Silicon wafers were first cleaned using buffered oxide etch and then spin-coated with the photoresists using manufacturer specifications to achieve 10-μm layers of each photoresist. The resists were then cured using UV light exposure through photomasks designed in CAD software and printed by CAD/Art Services Inc. (Bandon, Oregon), again following photoresist manufacturer specifications. Subsequently, wafers were baked and the uncured photoresist was dissolved. After curing the SPR coat, the features were then baked further to produce a rounded profile. After both wafers were complete, a protective coat of silane was applied by vapor deposition.

To cast and manufacture the two-layer device, the top wafer was coated with a 5-mm-thick layer of degassed 10:1 PDMS-to-curing agent ratio (Sylgard 184; Dow Corning). For the bottom layer, a 20:1 mixture was prepared and spin-coated to create a 100-μm-thick layer. The two layers were partially baked for 45 min at 70°C. The top layer was then cut and separated from the wafer, and holes were punched from the feature side using a sharpened 20-gauge blunt-tip needle to make external connections to the control valve lines, then aligned and laminated onto the bottom layer. The stacked layers were baked together for 1.5 h at 70°C and allowed to cool. The laminated layers were then cut out and the remaining ports were punched to make external connections with the channels.

Next, the cut and punched PDMS devices were bonded to 24 × 50 mm glass coverslips (#1.5). The PDMS was cleaned with transparent adhesive tape (Magic Tape; Scotch) followed by rinsing with (in order) isopropanol, methanol, and Millipore-filtered water, air-drying between each rinse. The glass was rinsed with acetone, isopropanol, methanol, and Millipore-filtered water, air-drying between each rinse. The PDMS device was tape-cleaned an additional time before it was placed with the coverslip in a plasma bonding oven (Harrick Plasma). After 1 min of exposure to plasma under vacuum, the device was laminated to the coverslip and then baked on an 80°C hotplate for 15 min to establish a covalent bond. Devices were stored at room temperature and used within 24 h.

## Microfluidic gradient experiment

For microfluidic chemotaxis assays, cells were streaked from frozen stocks onto LB agar plates. A single colony was picked and grown overnight to saturation in M9–glycerol medium plus the desired amount of IPTG, arabinose, ampicillin, and chloramphenicol, as appropriate. The OD of this overnight culture was measured and subcultured with shaking at 30°C for 4.5 generations to exponential phase ($OD_{600} \approx 0.15$) in 12–15 ml fresh M9G media plus the desired amount of IPTG, arabinose, and antibiotics as appropriate. The subculture was split evenly into two 15-ml Falcon tubes and spun for 5 min at 956 $g$ in a benchtop centrifuge. The supernatant was decanted, and the cells were gently resuspended in the residual liquid by rocking and flicking the tube. The resuspended pellets were then combined and transferred to a 1.5-ml Eppendorf tube. Cells were then washed three times in CB. Washing was done by centrifuging the tube for 3 min at 1,700 $g$ removing the supernatant with a pipette, gently adding 1 ml CB, and then resuspending the pellet by rocking and flicking. After washing, cells were resuspended in 0.5 ml CB, and diluted, if necessary, to an $OD_{600}$ of 0.8–1.5 for experiments with an attractant gradient or 2–3 for experiments without attractant.

Immediately before setting up the device, it was injected with 5 μl 10% w/v benzophenone in acetone. This organic photoinitiator permeated the PDMS and ensured robust PEG hydrogel polymerization (described below) at the fluid–PDMS interface.

Sample lines were connected to reservoirs mounted on a manifold of manual stopcocks allowing for independent three-way selection between tank pressure regulated at tunable 0–3 psi of compressed nitrogen ("on"); atmospheric pressure ("off"); and sealed (typically not used). The lines were primed and reservoirs filled by introducing negative pressure into the manifold through an auxiliary port and subsequently connected to the PDMS device via stainless steel 20-gauge connector blunt stubs. The valve lines were similarly set up, except all lines were filled halfway with water, no reservoirs were used, and the manifold had computer-controlled solenoid valves. Tank pressure to the valve manifold was regulated to 25 psi.

Hydrogel walls were created by polymerization of photoreactive derivative of poly(ethylene glycol) with diacrylate groups (PEGDA) using UV light, which was delivered from a high-pressure mercury lamp. The chamber was first flooded with wall solution (10% v/v PEGDA, 700 average molecular weight; 0.05% w/v LAP photoinitiator; 1 ng/ml resorufin to visualize solution) introduced through one of the sample lines. The walls were formed by engaging the microscope pinhole aperture for the light source (100-μm-diameter exposed area using a 10× objective), and exposing a stripe of PEGDA (in at least two passes) at the source and sink side of the observation chamber. The chamber was then flushed with CB from a different sample line until no resorufin dye remained (~40–60 min).

Source and sink solutions were continuously circulated on either side of the observation chamber through channels that were separated by the hydrogel walls so that diffusion was possible without cross flow. In both the gradient and no-gradient experiments, the sink solution was CB. In the gradient case, a solution of 1 mM MeAsp and 10 μM fluorescein in CB was used as the source, whereas in the no-gradient case, CB was used. To establish the gradient, all valves isolating the observation chamber from the rest of the device were engaged except for one on the sink side, which was left open to replace fluid lost to evaporation. This had no measurable effect on the shape of the gradient, but failure to perform this step resulted in the formation of negative pressure in the observation chamber, which could cause unexpected behavior

during cell loading. The gradient was allowed to equilibrate for 6.5 h before cell loading.

Cells were loaded into one of the sample line reservoirs and cycled into a section of device separate from the main channel to avoid disturbing the gradient. The cell-retaining gate was then closed, and cells were flowed into a narrow strip of the sink side of the main channel behind the gate. All valves surrounding the main channel were then closed, and the experiment was started by simultaneously beginning the automated acquisition and opening the cell-retaining gate.

### Running experiment and data acquisition

A custom MATLAB script was used to control the automated stage (Prior) of the microscope (Nikon Ti-U) via the MicroManager interface (Edelstein *et al*, 2014). Starting at the gate, eight 1-min movies were sequentially acquired across the observation chamber using an EM-CCD camera (a 1,024 × 1,024 array of 13 × 13 μm pixels; Andor) or a scientific CMOS camera (2,048 × 2,048 array of 6.5 × 6.5 μm pixels, with 2 × 2 binning; Hamamatsu) at 8.78 frames/s through a 10× phase-contrast objective (Nikon CFI Plan Fluor, N.A. 0.30, W.D. 16.0 mm). After reaching the end of the observation chamber, acquisition started over at the gate. Overlapping regions were accounted for during data analysis. For the experiments without a gradient, four positions were used to observe the first 4.5 mm of the chamber. Before and after each movie, a fluorescence image was acquired by a LED illuminator (Lumencor SOLA light engine, Beaverton, OR) through the YFP block (Chroma 49003; Ex: ET500/20x, Em: ET535/30 m) to capture the fluorescein gradient. Fluorescein has approximately the same diffusion coefficient as MeAsp (Ahmed *et al*, 2010), making it a good indicator of the MeAsp gradient in the device.

### Single-cell tracking and run/tumble processing

To identify objects, the movies were background-subtracted by averaging over 6-s windows and subtracting that average from each frame in that window. We used the radial-symmetry-center method (Parthasarathy, 2012) to identify objects and tracked the motions of objects from frame to frame using the U-track software package (Jaqaman *et al*, 2008).

Cells were assumed to be in one of three possible states, "run", "tumble", or "intermediate". The intermediate state captured the entrance or exit of a cell from a tumble (Berg & Brown, 1972), or the minor changes in speed and direction that occur when a small fraction of flagella change to clockwise rotation (Turner *et al*, 2016). States were assigned based on a previously described clustering algorithm (Dufour *et al*, 2016).

### Analysis of trajectories

Movies were processed on the Omega cluster at the Yale High Performance Computing facility or on Dell workstations. To remove non-cell tracks (e.g., dust and spurious detection of microfluidic device features) and damaged cells, we filtered the tracks to have a mean speed (with tumbles) and mean run speed (excluding tumbles) between 5 and 60 μm/s, a maximum mean squared displacement of at least 30 μm$^2$, and a tumble bias between 0 and 0.5. We also filtered tracks on the estimated spatial standard

deviation provided by the detection algorithm (Parthasarathy, 2012). Large values of this parameter tended to be out-of-focus background noise. The threshold value of this parameter, which was sensitive to the exact focal plane of the experiment, was determined manually on a per-experiment basis.

Track length presented a particular problem. On the one hand, confidence in the estimate of tumble bias increases with the length of the track. On the other hand, discarding too many short tracks distorts the density estimate. This is because track length is inversely correlated with higher cell density, and cells are moving from a region of high cell density to a region of low cell density. We picked a minimum track length of 6 s, since this did not significantly distort our estimate of position versus time, and provided at least 53 frames to estimate a tumble bias. Consistent with this idea, increasing the minimum track duration to 20 s appeared to slightly increase the mean position of cells with lower tumble bias, but this effect did not significantly change the outcome (Appendix Fig S4).

Movies were subdivided into five spatial regions along the gradient axis to calculate the cell density across the movie. The mean position as a function of time was calculated by taking the average position of all cells detected in each sink-to-source scan ("sweep") of the microscope stage across the device. The time points were determined by the midpoint of each stage scan.

The total number of cells observed was estimated by assuming that a track observed for the entire $n$ frames of a movie represented one cell. Thus, one frame of observation was counted as $1/n$th of a cell, and the total number of cells was calculated by multiplying the total number of objects observed over all frames of all movies by $1/n$.

### Quantifying the gradient profile

The shape of the MeAsp gradient was reconstructed from the fluorescein images by first normalizing each image to a flat-field image acquired from a reference slide (Ted Pella, Redding, CA) or from a "blank" fluorescent image of the observation chamber before setting up the gradient (i.e., in the absence of fluorescein) to correct for systematic distortions in fluorescence illumination across the frame. Images from the sink and source loops (which were also flat-field-corrected) were used as reference values to determine the fluorescence intensity corresponding to 0 and 1 mM MeAsp, respectively. These values were used to determine the concentration of MeAsp at each position along the gradient. Images taken before and after each movie were averaged. The resulting intensity profile was averaged across the dimension orthogonal to the gradient to produce a 1D profile along the gradient dimension, and regions of overlap were trimmed. The resulting space–time surface of gradient concentration information was smoothed by fitting a fifth-order, two-dimensional (space and time) polynomial.

### Strain construction

The mCherry-tagged CheR gene was inserted into the chromosome by first recombining it into the readily recombining strain MG1655 and then transferring into a CheR-deletion mutant of the infrequently recombining RP437 strain using P1 phage transduction. Cells were grown in standard recipes for Luria broth (LB),

   

super-optimal broth (SOB), or SOB with catabolite repression (SOC) as specified. Plates were made with media plus 1.5% agar and antibiotics as specified. Gibson assembly (Gibson *et al*, 2009) was used to a create plasmid containing the recombination insert. The complete insert consisted of a mCherry::cheR sequence encoding an N-terminal fusion protein followed by an FRT-sequence-bounded (Cherepanov & Wackernagel, 1995) kanamycin resistance cassette. The source plasmid for the mCherry sequence was created by first codon-optimizing published sequences for expression in *E. coli* and then *de novo* synthesizing plasmids (Invitrogen) containing these sequences within the pUC19 cloning vector sequence. The sequence for cheR was PCR-amplified from the RP437 genome. The FRT-kanR-FRT cassette was PCR-amplified from pCP15 (Cherepanov & Wackernagel, 1995). The vector backbone for the final plasmid construct was pUC19.

The assembled recombination insert was PCR-amplified from the plasmid with primers containing homology for the region following chromosomal pLac in MG1655. Linear insert fragments were gel-purified. They were PCR-amplified again from the fragment using the same template and gel-purified again. Before transforming with the fragment, MG1655 cells were first transformed with pKD46 (Datsenko & Wanner, 2000) encoding a lambda-red recombinase cassette and selected on LB with ampicillin (100 μg/ml) plates at 30°C. Next, an overnight culture was prepared in 5 ml SOC with 100 μg/ml ampicillin. Overnight culture was diluted 1:500 into 5 ml SOB with 100 μg/ml ampicillin and either 1 mM or no arabinose and grown to OD 0.6 (about 3 h). The subculture was centrifuged at 3,800 *g* for 7.5 min at 4°C. After aspirating the supernatant, the pellet was washed with 1 ml 10% glycerol in a 4°C temperature-controlled room three times by centrifugation for 3 min at 3,800 *g*, then resuspended in 50 μl 10% glycerol. To this suspension, 1 μg of recombination insert fragment DNA was added and electroporated, followed by immediate recovery in 1 ml SOC at 37°C for 2 h. The culture was then washed twice with 1 ml M63 salts by centrifugation for 3 min at 3,800 *g*, then resuspended in 100 μl of M63 salts and spread on LB agar with 50 μg/ml kanamycin. Plates were incubated overnight at 43°C to remove pKD46. Colonies were streaked out twice on LB agar with 50 μg/ml kanamycin to purify and screened for ampicillin sensitivity. The insertion site was PCR-amplified from a genomic DNA prep and verified by sequencing.

To create the P1 phage donor lysate, overnight saturated culture of the donor strain in LB was first diluted 1:100 into 3 ml transduction growth medium (TG; LB with 5 mM CaCl$_2$ and 0.2% w/v glucose). The subculture was grown 30 min at 37°C, and then, 75 μl P1 phage stock was added. After a 3-h incubation, 10 μl of chloroform was added. The sample was then pelleted and the aqueous supernatant was saved and stored at 4°C.

In order to transduce mutations from donor lysate into the recipient strain, the overnight saturated culture in LB of the recipient strain was diluted 1:100 into 10 ml TG until late exponential phase at 37°C. The cells were pelleted and resuspended in 2.5 ml transduction reaction medium (TR; LB with 5 mM CaCl$_2$ and 100 mM MgSO$_4$). A dilution series was prepared from 1:1:0 to 1:0:1 of recipient cells, TR, and donor lysate in a total volume of 200 μl. Transduction reactions were incubated without shaking at 37°C for 30 min. Cells were grown in 1 ml of LB combined with 200 μl of 1 M sodium citrate pH 5.5 for 1 h at 37°C with shaking. Next, cells

were pelleted and resuspended in 100 μl transduction selection medium (TS; LB with 20 mM sodium citrate) and plated on TS agar with 25 μg/ml kanamycin and grown overnight at 37°C. Large colonies were selected and colony-purified twice on TS agar with 25 μg/ml kanamycin.

Following transduction, the antibiotic marker was removed via FRT excision by first transforming with pCP20 (Cherepanov & Wackernagel, 1995) and selecting on LB with 100 μg/ml Amp at 30°C overnight to remove the FRT-bounded region. After re-streaking on LB with no antibiotic, cells were grown overnight at 43°C to remove pCP20. Sensitivity to both kanamycin and ampicillin was verified, and the insertion site was sequence-verified.

### Simulations

Simulations were performed using a standard model of the chemotaxis pathway and previously described simulation methods (Sneddon *et al*, 2012; Frankel *et al*, 2014). Population diversity was created using a noisy gene expression model (based on Løvdok *et al*, 2009), which takes into account the effect of promoter sequences, ribosome binding sites, and operon structure to generate individual cells each with different numbers of the chemotaxis proteins (CheRBYZAW) and receptors (Tar, Tsr; Frankel *et al*, 2014). The gene expression, molecular pathway, and dynamics-based simulation were unchanged from our previous work (Frankel *et al*, 2014). We note changes to specific parameter values below. Our previous model assumed that cells had one flagellum per cell, but to make the model more realistic, we added a model of multiple interacting flagella (Sneddon *et al*, 2012) and used five flagella per cell with a 3 flagella minimum bundle size for each cell.

Most parameters were unchanged from our previous simulation paper, but some parameters of the model were adjusted to achieve better agreement with the experimental observations in this work. The number of Tar receptors per assistance neighborhood was changed to 6, which is consistent with previously reported values (Keymer *et al*, 2006; Mello & Tu, 2007; Shimizu *et al*, 2010). Following the inclusion of the multiple flagella model, the parameter controlling the amount of extrinsic noise in the gene expression model, $\omega$, was reduced from 0.26 to 0.08, and the autophosphorylation rate of CheA was raised from 12.2 to 12.35 in order to produce better agreement with the experimental tumble bias distribution. A Gaussian distribution of run speed was introduced in the model with mean and variance chosen to closely match the given data being modeled. Run speed was not correlated with the other parameters. When modeling the populations with inducible CheR expression, the parameter representing the mean level of CheR expression in the noisy gene expression model was adjusted until good agreement with the experimental tumble bias distribution was observed. See the legend to Fig EV3 for the speed distribution parameters and the mean number of CheR molecules used in each simulation.

We found that cells in the device without a gate or attractant gradient had a mean run-to-run change in angle of 81° (90° indicates no angular persistence). This is substantially greater than the previous observation of 62° (Berg & Brown, 1972) and could be due to differences in our experimental conditions. We used different growth media, different motility media, and a different strain. In addition, the observations of Berg and Brown (1972) were

performed in a 3D environment, whereas our cells were constrained to a pseudo-2D environment with a depth of 10 μm. Since simulations showed that such a small persistence did not affect cell performance, we did not include directional persistence in the model.

In order to simulate cells in the microfluidic environment, we introduced cell–boundary collision rules to the simulation. Experimentally, cells are observed to slow down slightly when colliding with a boundary, and then follow the boundary for a short time after the collision. Thus, when cells crossed a boundary in the simulation, their centroid was repositioned to the nearest boundary at the end of the time step and their headings redirected to lie in the boundary plane. This had the effect of generally causing them to proceed along the boundary until redirected by diffusion or a tumble, similar to observed cell behavior in microfluidics. The gradient used in the simulation was exactly the function derived from experimental data (see "Quantifying the gradient profile"). Cells in the simulation were initialized in a region corresponding to the area behind the gate in the experiment, and the simulation environment dimensions were also matched to the experiment. Each simulation had 16,000 trajectories.

In order to analyze the simulations and the experiments in the same way, a pseudo-microscope procedure was applied to the simulation data. A series of "movies" with the same time and space boundaries as those in the experiment were created by logically masking the simulated cell trajectories according to these bounds. Paralleling the experimental acquisition, the portions of cell trajectories that fell within each mask were extracted and treated as new, independent trajectories containing information about position and run state at each time point. This was repeated along the simulation environment and over simulation time following the position and time data from the microscope. Downstream analysis of the simulation data was handled in the same manner as the experimental data.

**Single-cell fluorescence measurements**

Cells were prepared as described in "Microfluidic gradient experiment" except that cells were resuspended in the residual buffer after the last wash. To create agar pads, two identical glass slides wrapped with a single layer of masking tape were placed on either side of a standard glass slide. 10 ml of 1% agarose was prepared and allowed to cool for 5 min. 60 μl of agarose was added to the middle slide, and a fourth slide was placed on top perpendicular to it. After the agarose solidified, the top slide was carefully removed by sliding it off the bottom slide. 0.5–1 μl of concentrated cell solution was added to the pad, and a coverslip was placed on top. The edges were sealed using VALAP (1:1:1 Vaseline:lanolin:paraffin by weight). Phase-contrast and fluorescence (RFP; Chroma 49008; Ex: ET560/40×, Em: ET630/75 m) images were taken at 100× magnification in order to determine cell outlines and mCherry relative concentration, respectively. RFP-channel images of wild-type RP437 were used to determine autofluorescence, which was negligible (< 1% of signal). An RFP-channel image of a reference slide (Ted Pella, Inc.) was used for flat-field correction. Flat-field correction was done using Fiji (Schindelin *et al*, 2012) by first dividing the image of the reference slide by its mean. Then, each sample image was divided by this reference image. Flat-field-corrected images were analyzed using Microbe-Tracker (Sliusarenko *et al*, 2011), modified to work with Linux (source code available at https://github.com/nodice73/MicrobeTrac

ker; parameters used for analysis are saved as "alg4ecoli_AW.set" in this repository). Background correction was done within Microbe-Tracker. The meshes on each analyzed image were manually inspected and corrected if necessary. Fluorescence histograms were generated using the "mtHist.m" script.

**Expanded View** for this article is available online.

## Acknowledgements

We thank Robert Austin for discussion about the microfluidic design, the Stanford Microfluidics Foundry for protocols, Sandy Parkinson and Tom Shimizu for strains, and Paul Turner and Douglas Weibel for discussions. Many simulations and data analysis routines were run on the clusters maintained by the Yale Center for High Performance Computing. This study was supported by the National Institutes of Health grant 1R01GM106189, the Allen Distinguished Investigator Program (grant 11562) through The Paul G. Allen Frontiers Group, and the James S. McDonnell Foundation grant on Complexity.

## Author contributions

TE conceived and supervised the project. TE, YSD, NWF, and AJW designed the research. YSD designed the microfluidic device. YSD and AJW developed the computer control and analysis software. YSD and NWF constructed the strains. AJW and NWF performed the experiments. JFJ performed control experiments. AJW, NWF, YSD, and TE performed the data analysis. NWF, JL, and AJW performed the simulations. AJW, NWF, and TE wrote the manuscript. All the authors discussed the manuscript.

## Conflict of interest

The authors declare that they have no conflict of interest.

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
