## [Review Process File · Molecular Systems Biology]

Non-genetic diversity modulates population performance

Adam Waite, Nicholas Frankel, Yann Dufour, Ms. Jessica Johnston, Junjia Long and Thierry Emonet

Corresponding author: Thierry Emonet, Yale University

Review timeline:

Submission date:	27 April 2016
Editorial Decision:	31 May 2016
Revision received:	15 November 2016
Accepted:	16 November 2016

Editor: Thomas Lemberger

Transaction Report:

1st Editorial Decision

31 May 2016

Thank you again for submitting your work to Molecular Systems Biology. We have now heard back from two out of the three referees who agreed to evaluate your manuscript. Given that their recommendations are very similar, I prefer to make a decision now rather than delaying the process further. As you will see from the reports below, the referees find the topic of your study of potential interest. They raise, however, several points that should be convincingly addressed in revision. The recommendations provided by the reviewers are very clear in this regard.

REFeree REPORTS

Reviewer #1:

The manuscript by Waite et al. aims to understand the connection between non-genetic variability and population performance. The specific question is how biological function emerges as a coevolution of the distribution of phenotypes and the single-cell performance function. The authors decided to use *E. coli* chemotaxis as a model system since its biased random walk from 'runs' and 'tumbles' is well characterized. In particular, the tumble bias (probability to tumble) is used as a read-out of phenotype. To investigate performance the authors combine microfluidics, cell tracking, and simulations of swimming cells to show that rare phenotypes of low tumble bias strongly improve the population performance in terms of chemotactic drift up the gradient. This statement is formalized using Jensen's inequality, which says that the average of a convex function (chemotactic drift as a function of tumble bias) is larger than the function value of the average. By expressing a key adaptation enzyme (CheR) from an inducible plasmid, they show that the changes in tumble bias have the predicted effect on the drift. While it is well known that chemotactic performance or drift depends on tumble bias and level of adaptation enzymes, the manuscript formulates the

mapping between phenotype variation and population performance in an elegant and clear way. Also the variability of the single-cell trajectories has not been quantified in the past. The findings of the paper may play a role in predicting the time of an infection based on chemotactic variability. The paper is overall convincing and well written to interest a broad audience.

Major points:

What if a different readout of phenotype or performance would have been used? Trivially, if the 'coherence' of the swimming across the cell population is used as a performance measure, then a broad or long-tailed distribution of phenotype is overall bad performance. There might also be concave functions where the average of the population performance is below the performance of the average cell, e.g. anything related to 'precision'. Hence, it seems very subjective to choose one example which 'works'.

The trajectories are rather short (1 min). If trajectories are longer, will rotational diffusion not destroy good chemotaxis at low tumble bias? So low tumble bias is not universally good as this depends on length scale of interest.

Chemotactic performance also depends on gradient steepness, speed of adaptation, level of CheB enzyme, and level of CheY response regulator, just to name a few. However, the roles of these parameters are not discussed. Please explain chosen selection of what to consider and what not.

Are measured trajectories in 2D or 3D? Same question applies to simulations. Is tumble bias measured in a gradient, and if so, tumble bias should change along gradient. How is this taken care off?

An earlier model of the group, published in Dufour et al. (2014), does not account for the bias in angles after a tumble which may increase the drift at high tumble bias. Is this included in the current model?

Minor points:

In Fig. 1 B-D it is unclear what the 'distance past gate' means in terms of the schematic of the flow chamber in panel A. Please indicate 'distance' and 'gate' in this schematic.

Reviewer #2:

In this paper, the authors use the chemotactic behavior of *E. coli* as a model to investigate the effect of a broad distribution of phenotypes on the efficiency in performing a specific task by a population of clonal individuals. They use a cleverly designed microfluidic device and automated data collection and analysis to record the motion of thousands of *E. coli* cells in a channel with a pre-established gradient of chemoattractant. This allows them to correlate a specific phenotype (tumble bias) with the performance (distance travelled up the gradient), using both wild-type cells and a strain that expresses the methyltransferase CheR under the control of inducible promoter. They showed that both diffusive spreading in absence of a gradient and chemotactic drift were higher for low but non-zero tumble bias. They also performed simulations of the chemotactic behavior, in excellent agreement with the experimental results. They then used all their data to construct a phenotype to performance map, which was non-linear and convex, and concluded that highly performing outliers that correspond to very low tumble bias are important in bringing the average performance of the population above the performance of the average cell. Notably, this effect could be mathematically formalized using Jensen's inequality and it is likely to play a role in many biological systems.

This manuscript is insightful and technically solid and the use of *E. coli* chemotaxis as a model system to address effects of population heterogeneity and cell individuality has an excellent tradition. I am confident that this work will be of interest to a general readership of MSB. However, I do have several critical comments that need to be addressed before the manuscript is accepted for publication.

Major point:

Although it is clear that cell performance in a gradient depends on tumble bias in a non-trivial way, this dependence might be more complex than currently suggested by data presentation and at least partly indirect. Can the authors clearly distinguish direct dependence of performance on tumble bias from dependence on other bias-related behavioral parameters? In this context, titration of the adaptation enzyme CheR may not be an ideal way to regulate tumble bias, since levels of CheR similarly affect kinetics of adaptation, another key determinant of the chemotactic performance. Authors' own previous theoretical work clearly established the importance of the adaptation kinetics (i.e, short-term memory) in chemotactic behavior (Emonet & Cluzel, 2008; Frankel et al., 2014). If I am not mistaken in my interpretation, computer simulations shown in Frankel et al. (2014) suggest that changes in adaptation kinetics due to variation in the levels of adaptation enzymes have even stronger effects on performance than changes in tumble bias. Moreover, at low levels of CheR tumble bias might not only be low but also highly variable in time, which might itself increase efficiency of chemotactic movement (Emonet & Cluzel, 2008).

Although the main message of the story will remain valid even if the correlation between performance and tumble bias is indirect, I think that current focus on tumble bias might be potentially misleading. However, this issue could be relatively easily resolved by titrating the response regulator CheY, because levels of CheY should tune mean tumble bias without affecting the adaptation kinetics or slow methylation-dependent bias fluctuations. I do not ask for redoing all experiments at varying levels of CheY, but I believe that an experiment that confirms that the same (or similar) performance-phenotype relation is observed when tumble bias is controlled by CheY levels would be important for data interpretation.

Minor points:

- Figure 3D: Although for each induction level of CheR taken separately the performance is ordered according to the tumble bias, it appears that the performance is significantly different for a fixed range of tumble bias depending on the induction level. Could it be related to effects of CheR on parameters other than tumble bias (see comment above)?

- For both WT and 10uM IPTG induced CheR - and also for DcheY strain as a control - it would be interesting to show the distributions of distances past the gate for each subpopulation with a given tumble bias, at least at a given time point (e.g. 20 min). Are they self-similar, or is the population spreading inherently different for low tumble biases? Only average travelled distances are shown so far.

- Materials and Methods, subsection on data acquisition (Line 378 and following): As far as I understand, the whole channel seems to be scanned sequentially using 1 min long movies, the whole scan being roughly 10 minutes long. Is it actually the case? This should be described more clearly (e.g. how many areas, are they overlapping). Also, the pixel size of the camera is missing.

- Materials and Methods, line 385: Please give the characteristics of the filters of the YFP block

- Line 419: typing mistake "track length is is inversely proportional... "

Continued on next page.

We would like to thank the reviewers for their thoughtful and constructive comments. We have addressed each comment/concern in a point-by-point manner below. Text in the manuscript that has been altered or added to address these issues has been colored blue.

Reviewer #1:

The manuscript by Waite et al. aims to understand the connection between non-genetic variability and population performance. The specific question is how biological function emerges as a coevolution of the distribution of phenotypes and the single-cell performance function. The authors decided to use *E. coli* chemotaxis as a model system since its biased random walk from 'runs' and 'tumbles' is well characterized. In particular, the tumble bias (probability to tumble) is used as a read-out of phenotype. To investigate performance the authors combine microfluidics, cell tracking, and simulations of swimming cells to show that rare phenotypes of low tumble bias strongly improve the population performance in terms of chemotactic drift up the gradient. This statement is formalized using Jensen's inequality, which says that the average of a convex function (chemotactic drift as a function of tumble bias) is larger than the function value of the average. By expressing a key adaptation enzyme (CheR) from an inducible plasmid, they show that the changes in tumble bias have the predicted effect on the drift. While it is well known that chemotactic performance or drift depends on tumble bias and level of adaptation enzymes, the manuscript formulates the mapping between phenotype variation and population performance in an elegant and clear way. Also the variability of the single-cell trajectories has not been quantified in the past. The findings of the paper may play a role in predicting the time of an infection based on chemotactic variability. The paper is overall convincing and well written to interest a broad audience.

Major points:

What if a different readout of phenotype or performance would have been used? Trivially, if the 'coherence' of the swimming across the cell population is used as a performance measure, then a broad or long-tailed distribution of phenotype is overall bad performance. There might also be concave functions where the average of the population performance is below the performance of the average cell, e.g. anything related to 'precision'. Hence, it seems very subjective to choose one example which 'works'.

The reviewer is right to point out that we focused on one metric to evaluate cell performance. We want to clarify that we did not choose this metric because it works. We were interested in chemotactic ability, and therefore we thought the most natural metric of performance would be how quickly each phenotype is able to climb up a gradient of attractant. A main point of the paper is to demonstrate how diversity can influence population function, not to show that diversity is necessarily beneficial. We fully agree with the reviewer that the paper would benefit from a broader discussion of how the phenotype-to-performance map of different tasks (such as those that capture the concept of "precision") might display different types of nonlinearities (concave and convex), and how when the map is concave, the average performance of the population should be lower than that of the average phenotype. We have added this text to the Discussion section.

The trajectories are rather short (1 min). If trajectories are longer, will rotational diffusion not destroy good chemotaxis at low tumble bias? So low tumble bias is not universally good as this depends on length scale of interest.

The characteristic time scale of rotational diffusion for a $\sim 1 \mu\text{m}$ bacterium is about 8 sec (Dufour et al., 2014). Recently, we experimentally confirmed that this value is at most 10 seconds (Dufour et al., 2016). So, an upper bound for track duration of 1 min is long enough to capture the effect of rotational diffusion. To check the effect of our minimum trajectory length of 6 seconds, we increased the minimum trajectory duration to 20 seconds. This did not destroy the performance of low tumble bias cells but instead slightly increased it (Appendix Fig. S4) for reasons that we discuss in the *Analysis of trajectories* section of the *Materials and Methods*. Therefore, the observed high performance of low tumble bias cells is robust to these changes in quantification parameters. We also show experimentally that, when tumble bias goes to zero, performance is degraded in (Fig. 3F) as predicted theoretically (Dufour et al., 2014), highlighting that the reviewer's hypothesized performance decrease is reached in our device, but only at zero tumble bias.

Chemotactic performance also depends on gradient steepness, speed of adaptation, level of CheB enzyme, and level of CheY response regulator, just to name a few. However, the roles of these parameters are not discussed. Please explain chosen selection of what to consider and what not.

The reviewer rightfully points out that we did not sufficiently discuss our rationale for our choice of parameters. We have added text in various parts of the Introduction and Results sections to address this concern. In addition, we have run an entirely new set of experiments where we varied tumble bias by manipulating the expression levels of CheY and CheZ instead of CheR and got similar results (Fig. EV5), showing that, at least in our experimental conditions, performance is dominated by tumble bias and not adaptation time (see also answer to Reviewer #2).

As for investigating the effect of gradient steepness, our device was designed to generate linear gradients and the dimensions of the device reflect compromises between various experimental constraints. For example, a longer device would produce a shallower gradient and provide more time to observe cells, but the gradient would take much longer to equilibrate when setting up the experiment. A shorter device would provide a steeper gradient and shorter setup times but would reduce the amount of time we would be able to observe the cells swimming before they reached the top of the gradient.

Are measured trajectories in 2D or 3D? Same question applies to simulations.

In the experiment, cells swim in 3D (the depth of the observation chamber is $10 \mu\text{m}$, which is ~ 10 times the size of a cell) but are tracked in 2D. We treat the simulation exactly like the experiment, so there the cells also swim in 3D constrained by boundaries in the same dimensions as the chamber but are tracked in 2D.

Is tumble bias measured in a gradient, and if so, tumble bias should change along gradient. How is this taken care off?

The reviewer is correct; the observed tumble bias of a cell will drop relative to its unstimulated tumble bias when that cell senses a gradient of attractant. The magnitude of the drop is a function of gradient steepness (Dufour et al., 2014). Since *E. coli* sense the logarithm of the gradient, the linear gradient in our device is perceived as being steepest at the start and shallowest at the end.

We cannot directly measure the unstimulated versus observed tumble bias in the same cells in our experiment setup. So, we performed a control simulation using an experimentally-derived gradient where all the cells were initialized to be identical and had a known unstimulated tumble bias of 0.23.

We then plotted the average difference between the observed and unstimulated tumble bias as a function of position in the chamber.

As expected, the difference is greatest at the beginning of the gradient where the perceived gradient is steepest (Appendix Fig. S3A). However, this rapidly falls to a small, constant difference (-0.004) within the first millimeter of the observation chamber. The simulation also suggests that the difference between stimulated and unstimulated tumble bias cannot account for the observed differences in performance (Appendix Fig. S3B).

We have added this point to the main text of the manuscript.

An earlier model of the group, published in Dufour et al. (2014), does not account for the bias in angles after a tumble which may increase the drift at high tumble bias. Is this included in the current model?

Initially, we did not include persistence in our model. We measured the average angular persistence of cells in experiments without a gradient or a gate and found that it was very small (mean change in angle was 81°). We then verified in simulations that including such small persistence did not change the results. The difference between our result and the commonly cited value of 62° (Brown and Berg, 1972. *Nature*.) could be due to differences in experimental conditions. Brown and Berg used a different strain, different growth media, different tracking media, tracked their cells in three dimensions, and their cells were not constrained to a $10\ \mu\text{m}$ depth as was the case in our observation chamber.

We have added this information to the “Simulations” section of the Materials and Methods.

Minor points:

In Fig. 1 B-D it is unclear what the 'distance past gate' means in terms of the schematic of the flow chamber in panel A. Please indicate 'distance' and 'gate' in this schematic.

We have updated Fig. 2A to specify the location of the gate.

Reviewer #2:

In this paper, the authors use the chemotactic behavior of *E. coli* as a model to investigate the effect of a broad distribution of phenotypes on the efficiency in performing a specific task by a population of clonal individuals. They use a cleverly designed microfluidic device and automated data collection and analysis to record the motion of thousands of *E. coli* cells in a channel with a pre-established gradient of chemoattractant. This allows them to correlate a specific phenotype (tumble bias) with the performance (distance travelled up the gradient), using both wild-type cells and a strain that expresses the methyltransferase CheR under the control of inducible promoter. They showed that both diffusive spreading in absence of a gradient and chemotactic drift were higher for low but non-zero tumble bias. They also performed simulations of the chemotactic behavior, in excellent agreement with the experimental results. They then used all their data to construct a phenotype to performance map, which was non-linear and convex, and concluded that highly performing outliers that correspond to very low tumble bias are important in bringing the average performance of the population above the performance of the average cell. Notably, this effect could be mathematically formalized using Jensen's inequality and it is likely to play a role in many biological systems.

This manuscript is insightful and technically solid and the use of *E. coli* chemotaxis as a model system

to address effects of population heterogeneity and cell individuality has an excellent tradition. I am confident that this work will be of interest to a general readership of MSB. However, I do have several critical comments that need to be addressed before the manuscript is accepted for publication.

Major point:

Although it is clear that cell performance in a gradient depends on tumble bias in a non-trivial way, this dependence might be more complex than currently suggested by data presentation and at least partly indirect. Can the authors clearly distinguish direct dependence of performance on tumble bias from dependence on other bias-related behavioral parameters? In this context, titration of the adaptation enzyme CheR may not be an ideal way to regulate tumble bias, since levels of CheR similarly affect kinetics of adaptation, another key determinant of the chemotactic performance.

We agree with the reviewer's comment and to address it we have done an entirely new set of experiments where, instead of modifying CheR levels, we manipulated CheY/CheZ levels. See also answer to the reviewer's comment about CheY below.

Authors' own previous theoretical work clearly established the importance of the adaptation kinetics (i.e, short-term memory) in chemotactic behavior (Emonet & Cluzel, 2008; Frankel et al., 2014). If I am not mistaken in my interpretation, computer simulations shown in Frankel et al. (2014) suggest that changes in adaptation kinetics due to variation in the levels of adaptation enzymes have even stronger effects on performance than changes in tumble bias. Moreover, at low levels of CheR tumble bias might not only be low but also highly variable in time, which might itself increase efficiency of chemotactic movement (Emonet & Cluzel, 2008).

It is correct that at low levels of CheR expression fluctuations in the methylation-demethylation kinetics become larger. For that reason our simulations do include noise in the methylation-demethylation reactions.

Regarding the effect of changing adaptation time versus that of changing tumble bias, the situation is complicated by the fact that in wild-type cells adaptation time and tumble bias are inversely correlated, as shown experimentally in single cells (Park et al., Nature 2010). This correlation is also maintained in the model used for our simulations (Frankel et al. 2014). However, we did examine how these two effects affect performance independently of each other in another recent theoretical study (Dufour et al., 2014). We found that, in shallow gradients, adaptation time only starts to matter for very low tumble bias cells (see in particular Figure 2A from Dufour et al., 2014) . Thus, for most of the range of tumble bias explored in our experiments, the dominant factor that effects performance is predicted to be tumble bias.

Although the main message of the story will remain valid even if the correlation between performance and tumble bias is indirect, I think that current focus on tumble bias might be potentially misleading. However, this issue could be relatively easily resolved by titrating the response regulator CheY, because levels of CheY should tune mean tumble bias without affecting the adaptation kinetics or slow methylation-dependent bias fluctuations. I do not ask for redoing all experiments at varying levels of CheY, but I believe that an experiment that confirms that the same (or similar) performance-phenotype relation is observed when tumble bias is controlled by CheY levels would be important for data interpretation.

We agree that our focus on tumble bias as the only important chemotactic phenotype was not the best way to present our results, and that both tumble bias and adaptation time could contribute to

performance. In addition to making this more clear in the Introduction and Results sections (please see the response to reviewer 1, above), we followed the reviewer's suggestion and performed additional experiments in a strain where we could control the amount of CheY and CheZ (see Fig. EV5). This allowed us to experimentally test low tumble bias cells in a manner that did not also affect adaptation time in the way that changing CheR does. When CheY was expressed at low levels relative to CheZ, this strain also showed increased performance with decreasing tumble bias similar to low CheR expression. We could also see a nonlinear performance increase at low tumble bias at early time points in this strain. Thus, tumble bias does appear to be the dominant phenotypic parameter in determining performance in shallow gradients.

Minor points:

- Figure 3D: Although for each induction level of CheR taken separately the performance is ordered according to the tumble bias, it appears that the performance is significantly different for a fixed range of tumble bias depending on the induction level. Could it be related to effects of CheR on parameters other than tumble bias (see comment above)?

We believe the reviewer is referring to the fact that, in Figure 3D, the positions of the high and low CheR populations in tumble bias bin 0.05 – 0.15 (green) do not have overlapping error bars. We do not know the mechanistic basis for this discrepancy, but the difference is also visible in the spatial distributions of the populations (Fig. EV2). It is possible that this is related to effects of CheR on parameters others than tumble bias.

- For both WT and 10uM IPTG induced CheR - and also for DcheY strain as a control - it would be interesting to show the distributions of distances past the gate for each subpopulation with a given tumble bias, at least at a given time point (e.g. 20 min). Are they self-similar, or is the population spreading inherently different for low tumble biases? Only average travelled distances are shown so far.

We followed the reviewer's suggestion and added a figure showing the spatial distribution of each tumble bias bin for each pass through the observation for wild-type cells, the high- and low-CheR experiments, as well as $\Delta cheY$ (we did not split this population into tumble bias bins, as this strain cannot tumble). This has been added as Fig. EV2 and is mentioned in the main text.

- Materials and Methods, subsection on data acquisition (Line 378 and following): As far as I understand, the whole channel seems to be scanned sequentially using 1 min long movies, the whole scan being roughly 10 minutes long. Is it actually the case? This should be described more clearly (e.g. how many areas, are they overlapping). Also, the pixel size of the camera is missing.

We apologize for not being clear. We have added a more detailed explanation of the acquisition procedure to the Materials and Methods section.

- Materials and Methods, line 385: Please give the characteristics of the filters of the YFP block

We have added this information to the Materials and Methods section.

- Line 419: typing mistake "track length is is inversely proportional... "

Thank you. We have fixed this mistake.

Thank you again for sending us your revised manuscript. We are now satisfied with the modifications made and I am pleased to inform you that your paper has been accepted for publication.

Corresponding Author Name: Thierry Emonet

Manuscript Number: MSB-16-7044